# Environmental Risk Factors for Childhood Acute Lymphoblastic Leukemia: An Umbrella Review

**DOI:** 10.3390/cancers14020382

**Published:** 2022-01-13

**Authors:** Felix M. Onyije, Ann Olsson, Dan Baaken, Friederike Erdmann, Martin Stanulla, Daniel Wollschläger, Joachim Schüz

**Affiliations:** 1Environment and Lifestyle Epidemiology Branch, International Agency for Research on Cancer (IARC/WHO), 150 Cours Albert Thomas, CEDEX 08, 69372 Lyon, France; olssona@iarc.fr (A.O.); friederike.erdmann@uni-mainz.de (F.E.); SchuzJ@iarc.fr (J.S.); 2Division of Childhood Cancer Epidemiology, Institute of Medical Biostatistics, Epidemiology and Informatics (IMBEI), University Medical Center Mainz, Langenbeckstraβe 1, 55131 Mainz, Germany; dabaaken@uni-mainz.de (D.B.); wollschlaeger@uni-mainz.de (D.W.); 3Pediatric Hematology and Oncology, Hannover Medical School, Carl-Neuberg-Str 1, 30625 Hannover, Germany; Stanulla.Martin@mh-hannover.de

**Keywords:** leukemia, childhood, radiation, ELF-MF, pesticides, petroleum, umbrella review

## Abstract

**Simple Summary:**

Leukemia is the most common type of cancer among children worldwide. The aim of this umbrella review was to provide an evidence-based summary of epidemiological studies on environmental risk factors and the risk of childhood acute lymphoblastic leukemia (ALL), by exposure window. Second aim was to assess the prevalence in the German population which determines the relevance on population level. Only low doses of ionizing radiation in early childhood and maternal exposure to general pesticides during pregnancy showed convincing evidence of an association with childhood ALL. Other risk factors vary in level of association. The results of this umbrella review should be interpreted with caution; as the evidence are mostly from case-control studies, where selection and recall bias are potential concerns.

**Abstract:**

Leukemia is the most common type of cancer among children and adolescents worldwide. The aim of this umbrella review was (1) to provide a synthesis of the environmental risk factors for the onset of childhood acute lymphoblastic leukemia (ALL) by exposure window, (2) evaluate their strength of evidence and magnitude of risk, and as an example (3) estimate the prevalence in the German population, which determines the relevance at the population level. Relevant systematic reviews and pooled analyses were identified and retrieved through PubMed, Web of Science databases and lists of references. Only two risk factors (low doses of ionizing radiation in early childhood and general pesticide exposure during maternal preconception/pregnancy) were convincingly associated with childhood ALL. Other risk factors including extremely low frequency electromagnetic field (ELF-MF), living in proximity to nuclear facilities, petroleum, benzene, solvent, and domestic paint exposure during early childhood, all showed some level of evidence of association. Maternal consumption of coffee (high consumption/>2 cups/day) and cola (high consumption) during pregnancy, paternal smoking during the pregnancy of the index child, maternal intake of fertility treatment, high birth weight (≥4000 g) and caesarean delivery were also found to have some level of evidence of association. Maternal folic acid and vitamins intake, breastfeeding (≥6 months) and day-care attendance, were inversely associated with childhood ALL with some evidence. The results of this umbrella review should be interpreted with caution; as the evidence stems almost exclusively from case-control studies, where selection and recall bias are potential concerns, and whether the empirically observed association reflect causal relationships remains an open question. Hence, improved exposure assessment methods including accurate and reliable measurement, probing questions and better interview techniques are required to establish causative risk factors of childhood leukemia, which is needed for the ultimate goal of primary prevention.

## 1. Introduction

Cancer is the most common cause for disease-related mortality in children in high-income countries [1,2]. Approximately 1850 children under the age of 15 are diagnosed with cancer in Germany every year [3]. With over 11 million children in this age group, this corresponds to an average annual age-standardized incidence of 17.3 new cases per 100,000 children [3]. Childhood leukemia accounts for approximately 27% of all childhood cancers in the United States, 30% in Ireland and France, 35% in Shanghai, China and 33% in Germany [4].

Acute lymphoblastic leukemia (ALL) is the most common type of childhood leukemia. More than 80% of ALL cases are classified as B-lineage ALL [5]. Regarding the development of ALL in general, it is hypothesized that a first initial genetic alteration occurs in-utero (“first hit”), which is followed by further postnatal alterations [6,7]. Exposure to higher levels of ionizing radiation (IR) is an established environmental risk factor for childhood cancer [8]. Evidence for this association comes from different sources: partly from studies of atomic bomb survivors in Hiroshima and Nagasaki [9,10,11,12,13,14], partly from a large number of studies on therapeutic and diagnostic use of IR in medical settings [15].

While treatment and survival from childhood ALL has remarkably improved over the past decades [3], survivors are yet at risk of a wide spectrum of somatic late effects, and adverse psychosocial and socioeconomic consequences during later life, including treatment-induced second malignancies [16]. Therefore, establishing primary preventive measures remains the central goal with identifying modifiable risk factors being essential, as only few risk factors have been established so far. Reviews published in the early 2000s provided an overview on childhood leukemia and cancer in general [17,18,19]. In a review by Schüz and Erdmann [8] on potential environmental risk factors, additional exposures are discussed and evaluated for the level of evidence for an association with childhood leukemia. These exposures include parental factors such as exposure to pesticides, diet, alcohol consumption, and smoking [8]. Another potential risk factor for childhood leukemia discussed by Schüz and Erdmann is exposure to extremely low frequency magnetic fields (ELF-MF).

Besides reviews on potential environmental risk factors, there are numerous of systematic reviews on single environmental factors and childhood cancer. In the evidence pyramid, systematic reviews are at the top. However, as more systematic reviews and meta-analyses are published, decision-makers need to integrate the accumulating evidence for a concise evaluation to answer their questions [20]. While systematic reviews can come to different results, umbrella reviews such as this, help to synthesize the evidence to give a consolidated overview.

The young age at diagnosis of childhood ALL suggests an inherited component and that factors prior to birth, including exposures in utero, as well as those in early childhood may be important risk determinants and therefore considered as relevant time windows. Here we present an umbrella review on environmental risk factors for childhood ALL. The aim of this umbrella review was (1) to provide a synthesis of the environmental risk factors for the onset of childhood ALL by exposure window, (2) evaluate their strength of evidence and magnitude of risk, and (3) estimate the prevalence in the German population to determine the relevance on population level in a specific setting nevertheless broadly representative for many high-income countries.

## 2. Materials and Methods

### 2.1. Umbrella Review Methods

An umbrella review is a review of previously published systematic reviews or meta-analyses and uses explicit, systematic methods (identification, selection, and appraisal of published systematic reviews) to collate and synthesize findings, with or without meta-analyses [21]. The current umbrella review was conducted in accordance with the Preferred Reporting Items for Systematic Reviews and Meta-Analyses (PRISMA) guidelines [22,23] and in line with an a priori protocol agreed on by all authors.

### 2.2. Eligibility Criteria

The eligible studies had to be systematic reviews including meta-analyses of observational studies (cohort and case-control studies) and/or pooled studies (meta-analyses based on the original data). They were included if they summarized relative risks (RR) as either incidence rate ratios (IRR) or odds ratios (OR) on environmental risk factors (all factors including parental lifestyle but not genetic) in relation to ALL in children. We selected the risk estimates for B-lineage ALL when available, thereafter total ALL where results were not presented by cell-type, and lastly leukemia (also including acute myeloid leukemia (AML)) when results were not presented by leukemia sub-type. To synthesize recent evidence, authors used articles published in the last two decades (2003 and 2021) with no language restriction. The choice of this timespan was to reduce the overlap of original studies, and to better reflect current exposure circumstances in view of prevention opportunities.

### 2.3. Information Sources

#### Search Strategy and Data Extraction

A search strategy with Population, Exposure, Comparator and Outcome (PECO) components was used to identify and retrieve articles through MEDLINE via PubMed, and Web of Science (WOS) databases. The PECO components included a list of key words and MeSH terms (MeSH terms for PubMed) such as Child*, Infant*, New-born, Adolescence, Teenage*, Youth*, Environmental Exposure, Occupational Exposure, Prenatal exposure, Maternal exposure, Residential exposure, Household exposure, Domestic exposure, Indoor exposure, Outdoor exposure, Radiation, Chemical exposure, Pesticides, Infection, Case-Control Stud*, Cohort, Cross-Sectional, Leukemia*, Leukaemia*, Acute lymphoblastic leukaemia and Acute lymphoblastic leukemia (asterisks represent wildcards). An initial search was performed in April 2021 and updated in October 2021. Snowball searches by screening reference lists, were used to identify additional articles. Final search results were exported, and automatically screened for duplicates in EndNote version X8.2, and later screened manually for accuracy. Following article screening and selection, we extracted from the included full texts; author and year of publication, study design, included number of studies, exposure, exposure window and summarized RR/OR with their respective 95% confidence intervals (CIs). We derived German prevalence data on environmental risk factors from relevant case-control studies in Germany, and surveys including parents and the general population [24,25,26,27,28,29,30,31,32,33,34,35,36,37,38,39,40,41].

### 2.4. Quality Assessment and Risk of Bias in Systematic Reviews

The included systematic reviews, but not pooled analyses, were subjected to a rigorous appraisal for methodological quality and risk of bias using A Measurement Tool to Assess Systematic Reviews (AMSTAR 2). The AMSTAR 2 quality assessment and risk of bias tool had 16 criteria. Each systematic review was assessed by verifying compliance of the criteria. For example, “did the research questions and inclusion criteria for the review include the components of PECO?” The question is answered with “yes” score 1, “partial yes” score 0.5, “no” score 0 and NA score 0. The total score for each systematic review was then converted to a percentage and rated accordingly. The ratings are High (100%), Moderate (≥75%), Low (≥50%), and Critically Low (<50%). We did exclude a systematic review if it was rated “critically low” because it cannot be relied on to provide an accurate and comprehensive summary of the available studies [42] (Appendix A: AMSTAR 2). The pooled analyses were not part of the quality assessment because these set of articles used mostly individual level data.

### 2.5. Evidence for Risk Factors of Childhood Leukemia

The strength of the association was evaluated using the summary RR/OR of the various meta-analyses and categorized as very strong (RR > 5), strong (RR > 2), moderate (RR > 1.5), modest (RR > 1.2), and weak (RR > 1). Strength of association, heterogeneity across studies, and number of studies were used to evaluate the strength of evidence. The evidence was categorized into “strong” (consistently strong or very strong risk estimates in quality systematic review and meta-analysis), “some” (consistent moderate risk estimates in quality systematic review and meta-analysis), “little” (consistent low risk estimates), “no” (consistency of no association) and “conflicting”. The category of “conflicting” was used when systematic reviews on the same subject matter came to different conclusions. The prevalence of risk factors of childhood ALL were estimated mostly from studies with only German data and few in combination with other countries. Where prevalence was quantified in a population it was categorised as high (>20%), common (>10–20%), moderate (>5–10%), modest (>2–5%), and rare (<2%).

## 3. Results

### 3.1. Search Strategy Outcome

Fifty-nine articles including 42 systematic reviews and meta-analyses, and 17 pooled analyses met the criteria and were included in the evaluation (Figure 1).

### 3.2. Quality Assessment and Bias

Out of 42 systematic reviews assessed for quality using AMSTAR 2, 21 had moderate, 14 low, and 7 critically low quality. We did not include the 7 of “critically low” quality in the decision for the risk factors since the outcome would not provide an accurate summary of the evidence (Appendix A).

### 3.3. Environmental Risk Factors

Out of 198 meta-analyses presented in the 59 articles (as several articles included more than one meta-analysis), 26 were meta-analyses of paternal environmental risk factors around the time of conception of the child (Table 1), while 78 were of maternal exposures during pregnancy (Table 2) and 94 of the analyses were on exposures occurring in early childhood (Table 3).

### 3.4. Paternal Preconception Exposure

Paternal preconception exposure to ELF-MF showed “no” evidence of association with childhood leukemia overall or by subtype, or when using alternative reference categories for the purpose of comparison with previous studies [43]. Talibov and co-authors estimated ELF-MF with 9723 childhood leukaemia cases and 17,099 controls of occupational data (using job exposure matrix (JEM)) from the Childhood Cancer and Leukemia International Consortium (CLIC). Increased paternal age, was found to have “little” evidence of association with childhood ALL. This was evaluated from the same CLIC data but by Petridou et al., who used 11 case control studies (7919 cases and 12,942 controls) recruited via interviews and five register-based control studies (8801 cases and 29,690 controls) through record linkage of population-based health registries [44]. Similarly, CLIC’s pooled analysis on paternal exposure to domestic paint (5 studies) and working as painter before conception (12 studies) was found to have “little” evidence of association with childhood B-lineage ALL [41,45]. The authors estimated the relationship in two different stages (within 1–3 months before conception and within the year before conception). On the other hand, there was “some” level of evidence for paternal exposure to pesticides in general before conception, herbicides (including molluscicide and rodenticide) as well as for household insecticide/miticide or fungicide use [46,47,48,49,50]. There was “little” evidence for pesticides used on pets. Paternal smoking during preconception was found to have “some” level of evidence from three systematic reviews [51,52,53] with a total of 29 original studies. These studies examined daily smoking, never and ever smoking during first trimester. Paternal alcohol consumption during preconception showed “no” evidence [54], the authors also compared never and ever alcohol drinkers, and there was no heterogeneity (I^2^ < 0.01%) between original studies. There was substantial overlap of original studies among the systematic reviews.

### 3.5. Maternal Preconception/Pregnancy Exposure

Maternal exposure to petroleum and solvents during preconception/pregnancy were found to have “some” evidence [55]. The CLIC study by Talibov et al. also examined the relationship between maternal exposure to ELF-MF (>0.1–≤0.2 and >0.2 µT) during pregnancy and the risk of childhood leukemia B-lineage ALL with the reference category of ≤0.1 µT. As they found no association, we judged it as not (“no”) having evidence for the association [43]. General pesticide exposure during preconception/pregnancy were found to have “strong” level of evidence. This is similar to home pesticides, herbicides, insecticides or fungicides and rodenticides (“some” evidence) but not pesticides used on pets which showed “little” evidence. Some earlier studies reported very high summary RR, [50,56] as compared to more recent studies [46,48,49]. Maternal alcohol consumption during pregnancy and the risk of childhood ALL was not associated with ALL in any of the analyses. The exposure categories were ever versus never drinkers, which may have been too crude [54,57]. High maternal consumption of coffee and cola during pregnancy were found to have “some” level of evidence of association with childhood ALL but not maternal consumption of tea (“no” evidence) [58,59,60]. Maternal smoking during pregnancy was not associated with risk of childhood ALL. In contrast, paternal smoking during the pregnancy of the index child had “little” evidence [51,53,55,61]. Maternal intake of fertility treatment was found to have “some” level of evidence of associated with childhood ALL. There was no significant heterogeneity across the original studies [62]. This was different for maternal intake of folic acid and vitamins known to maintain DNA integrity during pregnancy, as we found an inverse association with “some” level of evidence [63,64,65], although there was heterogeneity across original studies (in two meta-analyses of the systematic reviews). Infant high weight (≥4000 g) at birth was found to have “some” evidence of association with childhood ALL [66,67,68,69]. Meanwhile, infant preterm birth or low birth weight and birth order were not associated with childhood ALL [66,69,70,71]. Maternal age < 25 years was found to have “some” level of evidence of association with childhood leukemia but not increased maternal age [72]. There was significant heterogeneity across original studies [44]. Caesarean delivery during the birth of the index child and the risk of childhood ALL showed “some” evidence of association [73]. There was substantial overlap of original studies among all systematic reviews, and original studies were combined in the meta-analyses irrespective of the study design (case–control or cohort).

### 3.6. Postnatal Exposure

Exposure to high traffic density during childhood was found to have “little” evidence as a risk factor of childhood ALL, while nitrogen dioxide (traffic related air pollutant) resulted in “no” evidence.

However, the original studies for both traffic density and nitrogen dioxide showed significant heterogeneity across studies [74,75]. Exposure to benzene and living in proximity to a petrol station during childhood was found to have “some” evidence as risk factors of childhood ALL [75,76]. Exposure to ELF-MF during childhood was found to have “some” evidence as a risk factor of childhood leukemia. All meta-analyses had summary RR of >1.00 [77,78,79,80,81,82], including a recent publication by Seomun et al. [82]. On the contrary, Amoon et al. [81] reported no association in another recent publication. Their evaluation was based on a pooled analysis (individual level data) of four studies published between 2015 and 2017 [83,84,85,86]. These four studies were also in the systematic review of Seomun et al. [82]. Paternal smoking during childhood was found to have “little” evidence as a risk factor of childhood ALL but not maternal smoking in the same exposure window [51,52]. Exposure to breastfeeding (≥6 months) during childhood was found to have “some” evidence of being a protective (inverse association) factor of childhood ALL [87,88,89,90]. Similarly, day-care attendance and contact with any pets during childhood was also found to have “some” evidence of being a protective (inverse association) factor for childhood ALL [87,91]. Living on a farm during childhood was not associated with childhood ALL [72]. Exposure to domestic painting during childhood was found to have “some” evidence as a risk factor of childhood ALL [45]. Exposure to general pesticides during childhood were found to have “some” level of evidence as a risk factor of childhood ALL. Also similar to general pesticides are home pesticides, herbicides and rodenticides, but not pesticide used on pets, which showed “little” evidence. This was based on four systematic reviews and one pooled analysis [46,48,49,56,92]. Overlap of original studies exists among all systematic reviews for pesticides in this present umbrella review. Concerning low doses of ionising radiation during childhood, we found a “strong” level of evidence as a risk factor of childhood ALL, with summary RR > 2.00 [93]. For exposure to domestic radon during childhood, we found “conflicting evidence” from one available systematic review with two meta-analyses (case-control and cohort studies). The meta-analysis of case-control studies (8 studies; 10,803 cases and 16,202 controls) showed an elevated risk, while that of cohort studies (2 studies; 1428 cases) did not. This may have been due to lack of statistical power, crude exposure assessment or evening confounding factors in original studies [94]. Living near nuclear facilities during childhood was found to have “some” level evidence as a risk factor of childhood ALL [95].

### 3.7. Prevalence of Childhood ALL

The prevalence of risk factors for childhood leukemia in Germany was “high” for paternal smoking with children in the same house [24]. Exposure to nitrogen dioxide was also identified as “high” in Germany. Data of the German Microcensus 2019 [25], which is the largest annual household survey in Germany, showed that being born second also has a “high” prevalence with more than 44% of children having at least one sibling. In maternal intake of folic acid, we identified that approximately 81.70% of women in Germany are exposed to it during pregnancy as shown by Kersting et al., in a cross-sectional study in Germany including approximately 900 mothers. In the same survey it was shown that over 80% of the mothers in Germany at least tried breastfeeding [26,27]. Day-care attendance is also very frequent in Germany, especially in the age-group of 3–6 years with 57.71% attending day-care [28].

Exposure to maternal smoking during pregnancy [29], birth order 3 [30] and high birth weight were “rare” [31]. Furthermore, exposure to pesticides [32], maternal intake of fertility treatment [33], birth order 4 [30], proximity to nuclear facility and radiation were “modest”, while radon exposure during childhood, paternal and maternal alcohol intake [34,35,36] were moderate [37]. However, exposure to ELF-MF [38,39], low or high maternal and paternal age [40], paint [41], and birth order 5 and 6 were found to be “rare” [30]. However, these prevalence estimates have/are changing over time.

## 4. Discussion

In the present umbrella review we evaluated environmental risk factors in relation to childhood ALL by exposure time window, strength of evidence, and magnitude of risk in 196 meta-analyses from 35 systematic reviews and 17 pooled analyses. Risk factors associated with childhood ALL included paternal smoking during preconception and childhood, traffic density, benzene and living in proximity to petrol stations, nuclear facilities, ELF-MF and low doses of ionising radiation during childhood. Similarly, maternal fertility treatment, solvent and petroleum exposure, domestic painting general pesticides, coffee cola and diabetes during pregnancy were associated with childhood ALL. Also, caesarean section, birth weight and paternal age were associated with increased risk (“little” to “some”) of childhood ALL. Maternal intake of vitamins and folic acid, breastfeeding and day-care attendance during postnatal were inversely associated with childhood ALL. Maternal exposure to nitrogen dioxide, consumption of tea and parental alcohol consumption did not show evidence of association with childhood ALL. Likewise, living on a farm, contact with pets, birth order and gestational age were not associated with childhood ALL. Domestic radon showed “conflicting” evidence. We also estimated the prevalence of these exposures in Germany as a measure of their relevance, as occurrence of the risk factor in the population is pertinent for effective primary prevention.

Pesticides are a heterogeneous group of chemicals with over 5000 formulations used in preventing, controlling, or eliminating pests [96,97]. This fact may explain why the results for pesticides vary from “little” to “strong” in this umbrella review. These findings are consistent with a previous umbrella review [97] and are likely due to few data on specific active ingredients. In vitro studies show that insecticides have been implicated in leukemogenesis. For example, a human cell line exhibited metabolic changes consistent with a leukemogenic potential of organophosphorus insecticides such as isofenphos, diazinon and fenitrothion [98,99]. We found “some” level of evidence for an association between postnatal ELF-MF exposure and childhood ALL with higher risk estimates in the highest exposed categories [100]. The association between ELF-MF and childhood leukemia was found to be during childhood (postnatal), this is consistent with a study by Schüz and Erdmann [8]. Low doses of ionising radiation and living close to a nuclear facility showed “strong” and “some” level of evidence, respectively, in postnatal exposure assessment. Although there are limited studies, the association between radiation and childhood leukemia traces back to the Hiroshima and Nagasaki atomic bomb survivors of 1945 in Japan, where low doses of ionising radiation increased childhood leukemia [101]. The results for domestic radon were not consistent. It was observed that all original studies except for two (case control studies) found a negative effect estimate. Statistical power, bias, confounding factors as well crude estimation may have been the reason for the overall outcome. Benzene and living in proximity to petrol station were associated (“some” evidence) with childhood ALL in this present study. Benzene emanates from occupational settings to the general environment, exposing especially those living near the facilities [102]. The potential association between benzene exposure and childhood leukemia is consistent with previous studies [103,104,105,106]. The evaluation for domestic painting exposure (“little” evidence) was solely the finding of a pooled analysis from CLIC [45]. Other original studies had earlier identified domestic painting exposure as a potential risk factor for childhood ALL where risk increased with more rooms painted [107,108,109,110].

With respect to intrinsic factors such as caesarean section, maternal diabetes, paternal age and increased birth weight showed an association with childhood ALL, but not low birth weight and birth order. These findings are consistent with a previous study by Schüz and Erdmann [8]. In the category of lifestyle, behaviour and infection-related factors, breastfeeding, maternal intake of folic acid and vitamin, as well as day-care attendance were associated with reduced risk of childhood ALL as studies consistently showed an inverse association with risk.

Inconsistent epidemiologic studies have prompted a debate on the carcinogenicity of some risk factors of childhood leukemia [74,75,82]. This is mainly due to the challenges in measuring exposures accurately (information bias), participation and recall biases [17,111]. In our umbrella review, we identified associations which are empirical (statistical) associations but whether they are causal remains an open question for many of them. However, the review processes such as search strategies, quality assessment of methods, selection, and other inherent biases may have impacted the suggested associations [112]. For example, most of the systematic reviews combined different study design (case control and cohort studies, questionnaire-based and register-based) in their meta-analyses that produced the association. There was also inadequate characterization of exposures in some of the primary studies which may be due to limited availability or poor quality of historical data [113]. Hence, there is a need for increased understanding of occupational, environmental and biological measurement for research. This may form the basis for the use of improved tools to measure and estimate exposure levels more accurately [114].

For primary prevention and reduction of the prevalence of childhood ALL in Germany and other countries, it is pertinent to target modifiable risk factors which are not too rare. For example, paternal smoking, exposure to pesticides, nitrogen dioxide, proximity to nuclear facility and radiation among others, as identified in this umbrella review, are either modifiable or can be avoided. Also, usage of our identified prevalence data for the distribution of childhood cancer risk factors is clearly hampered by the lack of reliability and representativeness of the data. The sources were mainly literature sources from German governmental agencies (not peer reviewed) and for some risk factors data is outdated or simply not available.

The strengths of this umbrella review include the high number of pooled analyses, and the systematic reviews which were evaluated using AMSTAR 2, although the screening and selection but not evaluation of the outcome of the articles was carried out by one author. Another strength is the separation of different exposure windows.

Weaknesses include the overlap of original studies across systematic reviews and pooled analyses which could lead to “overlapping risk of bias” [115]. There is also lack of analyses by duration in most systematic reviews and pooled analyses. Furthermore, a limitation inherent in umbrella reviews is, that the evaluation was based on previously published meta-analyses and pooled analyses. The motivations for conducting those original analyses are not known but may be driven by convenience or to answer questions raised in very specific contexts, and not conducted with the aim of a balanced overview. This means that while for some risk factors there was an abundance of previous reviews while other factors may have been neglected. The major prominent example is the interplay between patterns of exposure to infections and the training of the child’s immune system, a biologically very plausible hypothesis [116], but in systematic reviews only addressed via perhaps weak proxies such as day-care attendance, living on a farm or birth order. Another limitation inherent in systematic reviews and thereby umbrella reviews is that bias affecting the original studies is not removed, but rather is exaggerated; so some of the associations seen could be due to bias as the majority of ALL case-control studies suffer from the same vulnerability to recall and selection bias. ELF-MF is an example where the epidemiological association was established more than 20 years ago but concerns about bias and the lack of biological plausibility of the association have precluded any conclusions on causality [8]. There was a general lack of prevalence data and no uniform pattern for the extraction. Some of the data from the various German websites were not primarily designed for childhood leukemia. For example, the data on paternal smoking during early childhood, were fathers with children in the same house, were not specific to children with leukemia.

## 5. Conclusions

In conclusion, exposure to low doses of ionizing radiation during childhood was “strongly” associated with childhood ALL as well as general pesticide exposure during pregnancy in several studies, but not all. The results of the present umbrella review should be interpreted with caution due to the potential of information and selection bias in the underlying original studies. Hence, improved assessment methods including accurate and reliable measurement, probing questions and better interview techniques as well as enabling or improving the possibility to utilize secondary data for research purposes that will lead to establishing causative risk factors of childhood leukemia, are urgently needed for the ultimate goal of identifying modifiable risk factors for primary prevention.

## Figures and Tables

**Figure 1 cancers-14-00382-f001:**
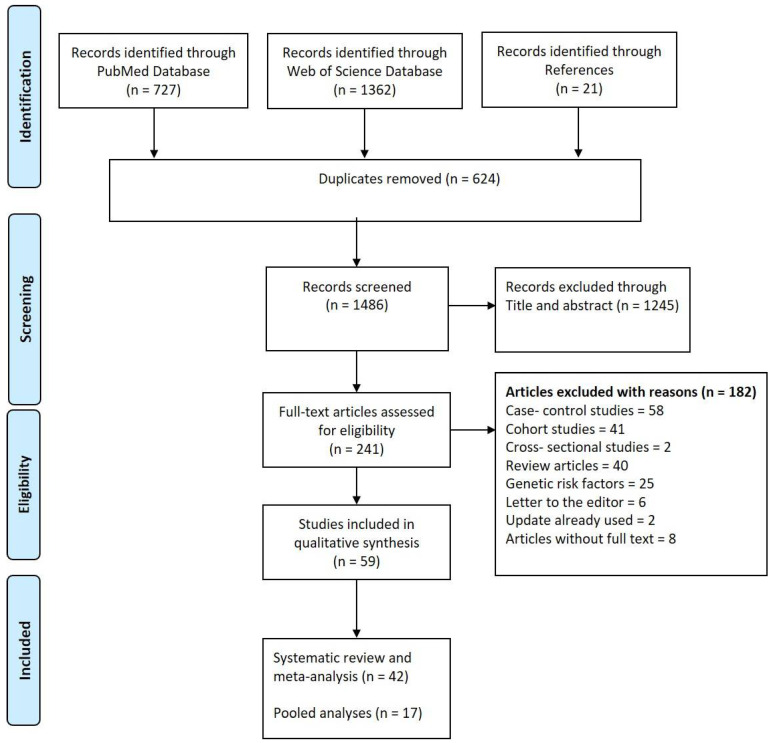
PRISMA flow chart describing the selection of included publications.

**Table 1 cancers-14-00382-t001:** Paternal preconception exposure to environmental factors in relation to childhood acute lymphoblastic leukemia in their offspring, including strength of evidence, prevalence of the risk factors in Germany, and magnitude of risk (RR) with 95% confidence intervals.

Authors	Study Design	Number of Study	Exposure Group	Exposure Type/Agent	* Evidence	Leukemia	^†^ Prevalence	^‡^ RR 95% CI
Talibov et al., 2019	Pooled analysis	11	Electromagnetic fields	>0.2 µT				1.09, 0.99–1.19
				>0.1–≤0.2 µT				0.93, 0.86–1.00
				>0.2–≤1 µT				1.04, 0.93–1.16
				>1 µT	No	B-lineage ALL	Rare	0.91, 0.62–1.31
Petridou et al., 2018	Pooled analysis	15	Intrinsic	Paternal age (increased age)	Little	ALL	Rare	1.05, 1.00–1.11
Karalexi et al., 2017	Systematic review	9	Lifestyle, behaviour, infection	Paternal alcohol	No	ALL	Moderate	1.10, 0.93–1.30
Chunxia et al., 2019	Systematic review	8	Lifestyle, behaviour, infection	Paternal smoking (Preconception)				1.15, 1.01–1.30
Liu et al. 2011	Systematic review	13		Paternal smoking				1.25, 1.08–1.46
Cao et al., 2020	Systematic review	8		Paternal smoking	Some	ALL	High	1.15, 1.04–1.27
Bailey et al., 2014b	Pooled analysis	12	Paint	Occupational painting				0.94, 0.76–1.15
Bailey et al., 2015b	Pooled analysis	5		Domestic painting within 1–3 months before conception				1.52, 1.25–1.86
		2		Domestic painting within the year before conception	Little	B-lineage ALL	Rare	1.01, 0.86–1.19
Van Maele-Fabry et al., 2019	Systematic review	4	Pesticides	General		ALL		1.30, 1.12–1.51
Bailey et al., 2014a	Pooled analysis	14		General-paternal occupational pesticide		B-lineage ALL		1.14, 0.85–1.54
Bailey et al., 2015a	Pooled analysis	2		General-occupational pest control treatments		B-lineage ALL		1.24, 1.03–1.50
Vinson et al., 2011	Systematic review	18		General		Leukemia		1.32, 1.20–1.46
Wigle et al., 2009	Systematic review	30		General	Some	Leukemia	Modest	1.09, 0.88–1.34
Bailey et al., 2015a	Pooled analysis	12	Pesticides	Home pesticide				1.41, 1.25–1.59
		5		Household insecticide/miticide				1.34, 1.19–1.51
		5		Insecticide or fungicide	Strong	B-lineage ALL	Modest	1.49, 1.14–1.95
		5		Pesticide used on pets	Little	B-lineage ALL	NA	1.17, 1.02–1.34
		5		Herbicide				1.23, 1.04–1.45
		5		Rodenticide				1.39, 1.10–1.76
		5		Molluscicide	Some	B-lineage ALL	Modest	1.06, 0.79–1.43

* Evidence category reflects those in the same rows by exposure type; ^†^ source of prevalence is different from RR data; ^‡^ RR also includes OR.

**Table 2 cancers-14-00382-t002:** Maternal preconception exposure to environmental factors in relation to childhood acute lymphoblastic leukemia in their offspring, including strength of evidence, prevalence of the risk factors in Germany, and magnitude of risk (RR) with 95% confidence intervals.

Authors	Study Design	Number of Study	Exposure Group	Exposure Type/Agent	* Evidence	Leukemia	^†^ Prevalence	^‡^ RR 95% CI
Talibov et al., 2019	Pooled analysis	11	Electromagnetic fields	>0.2				0.98, 0.85–1.12
				>0.1–≤0.2				0.95, 0.89–1.02
				>0.2	No	B-lineage ALL	Rare	0.96, 0.83–1.10
Karalexi et al., 2017	Systematic review	24	Lifestyle, behaviour, infection	Maternal alcohol		ALL		0.97, 0.85–1.11
	Systematic review	8		Maternal alcohol-moderate				1.13, 0.84–1.52
	Systematic review	8		Maternal alcohol-high				0.98, 0.71–1.36
Latino-Martel et al., 2010	Systematic review	11		Maternal alcohol	No	ALL	Moderate	1.10, 0.93–1.29
Thomopoulos et al., 2015	Systematic review	8	Lifestyle, behaviour, infection	Maternal coffee consumption (High)		ALL		1.43, 1.22–1.68
	Systematic review	9		Maternal coffee consumption (Low to moderate)				1.01, 0.90–1.13
Milne et al., 2018	Pooled analysis	7		Coffee > 2 cups/day		B-lineage ALL		1.28, 1.09–1.50
Cheng et al., 2014	Systematic review	5		Maternal coffee consumption (ever drinkers)	Some	ALL	NA	1.26, 1.05–1.50
Milne et al., 2018	Pooled analysis	5	Lifestyle, behaviour, infection	Maternal tea consumption >2 cups/day		B-lineage ALL		0.99, 0.80–1.24
Thomopoulos et al., 2015	Systematic review	6		Maternal tea consumption (High)		ALL		0.99, 0.84–1.18
	Systematic review	8		Maternal tea consumption (Low to moderate)	No	ALL	NA	0.90, 0.79–1.04
Thomopoulos et al., 2015	Systematic review	2	Lifestyle, behaviour, infection	Maternal cola consumption (High)		ALL		1.25, 0.95–1.66
		3		Maternal cola consumption (Low to moderate)				1.24, 1.03–1.49
Cheng et al., 2014	Systematic review	5		Maternal cola consumption (Low to moderate)		ALL		1.09, 0.91–1.31
		5		Maternal cola consumption (High)	Some		NA	1.65, 1.28–2.12
Chunxia et al., 2019	Systematic review	9	Lifestyle, behaviour, infection	Maternal smoking (preconception)		ALL	Common	1.05, 0.97–1.12
Chunxia et al., 2019	Systematic review	8		Paternal smoking during pregnancy		ALL	High	1.23, 0.99–1.53
Klimentopoulou et al., 2012	Systematic review	20		Maternal smoking during pregnancy		ALL	Common	1.03, 0.95–1.12
Chunxia et al., 2019	Systematic review	12		Maternal smoking during pregnancy		ALL	Common	0.97, 0.90–1.05
Zhou et al., 2014	Systematic review	18		Maternal smoking during pregnancy		ALL	Common	0.99, 0.96–109
Liu et al., 2011	Systematic review	8		Paternal smoking during pregnancy		ALL	High	1.24, 1.07–1.43
Cao et al., 2020	Systematic review	9		Paternal smoking during pregnancy	Little	ALL	Common	1.20, 1.12–1.28
Hargreave et al., 2014	Systematic review	11	Lifestyle, behaviour, infection	Fertility treatment	Some	Leukemia	Modest	1.65, 1.35–2.01
Metayer et al., 2014	Pooled analysis	8		Maternal Folic Acid		ALL		0.80, 0.71–0.89
Ismail et al., 2019	Systematic review	11		Maternal Folic Acid		ALL		0.75, 0.66–0.86
Metayer et al., 2014	Pooled analysis	12		Vitamin		ALL		0.85, 0.78–0.92
Goh et al., 2007	Systematic review	2		Multivitamin Supplementation	Some(inverse)	ALL	High	0.61, 0.50–0.74
Rudant et al., 2015	Pooled analysis	11	Intrinsic	Birth order ≥ 2		ALL		0.94, 0.88–1.00
				Birth order 2			High	0.95, 0.88–1.01
				Birth order 3			Common	0.95, 0.87–1.05
				Birth order 4			Modest	0.86, 0.73–1.00
				Birth order 5			Rare	0.92, 0.70–1.21
				Birth order ≥ 6	No		Rare	0.93, 0.68–1.29
Milne et al., 2013	Pooled analysis	12	Intrinsic	Weight (large-for-gestational-age)		ALL		1.21, 1.11–1.32
Hjalgrim et al., 2003	Systematic review	18		High birth weight = ≥4000 g		ALL		1.26, 1.17–1.37
Caughey et al., 2009	Systematic review	23		High birth weight		ALL		1.23, 1.15–1.32
Che et al., 2021	Systematic review	25		High birth weight	Some	ALL	Rare	1.28, 1.20–1.35
Wang et al., 2018	Systematic review	11	Intrinsic	Preterm birth		ALL		1.04, 0.97–1.11
Huang et al., 2016	Systematic review	8		Preterm Birth		ALL		1.04, 0.96–1.13
Caughey et al., 2009	Systematic review	10		Low birth weight		ALL		0.97, 0.81–1.16
Wang et al., 2018	Systematic review	10		Gestational age-post-term birth		ALL		1.03, 0.95–1.12
Che et al., 2021	Systematic review	27		Low birth weight	No	ALL	NA	0.83, 0.75–0.92
Marcotte et al., 2016	Pooled analysis	13	Intrinsic	Caesarean delivery		ALL		1.06, 0.99–1.13
				Prelabour caesarean delivery	Little		NA	1.23, 1.04–1.47
Petridou et al., 2018	Pooled analysis	15	Intrinsic	Maternal age (increased)		ALL		1.05, 1.00–1.08
Orsi et al., 2018				Maternal age > 35	Little	ALL	Rare	0.98, 0.89–1.08
	Pooled analysis	13		Maternal age < 25	Some	ALL	Rare	1.20, 1.11–1.29
Yan et al., 2020	Systematic review	9	Intrinsic	Maternal diabetes	Some	ALL	NA	1.44, 1.27–1.64
Bailey et al., 2014b	Pooled analysis	4	Paint	Occupational paint (Maternal)		B-lineage ALL		0.79, 0.36–1.71
Bailey et al., 2015b	Pooled analysis	8		Home paint-Any paint exposure		B-lineage ALL		1.14, 1.04–1.25
		8		Home paint-Mother used paint	Little		Rare	1.13, 0.95–1.33
Wigle et al., 2009	Systematic review	16	Pesticides	General		Leukemia		2.09, 1.51–2.88
Vinson et al., 2011	Systematic review	25		General		Leukemia		1.48, 1.26–1.75
Turner et al., 2010	Systematic review	5		General		ALL		2.04, 1.54–2.68
		4		General-Indoor exposure				1.86, 1.25–2.77
		5		General-Outdoor exposure				1.50, 0.98–2.32
Bailey et al., 2014a	Pooled analysis	12		General-maternal occupational		B-lineage ALL		1.04, 0.78–1.38
Bailey et al., 2015a	Pooled analysis	6		General-maternal professional pest control				1.19, 1.04–1.36
Van Maele-Fabry et al., 2019	Systematic review	5		General		ALL		1.39, 1.21–1.60
		5		General-Indoor exposure	Strong		Modest	1.27, 1.07–1.51
Bailey et al., 2015a	Pooled analysis	12	Pesticides	Home pesticide		B-lineage ALL		1.47, 1.35–1.61
		6	Pesticides	Household insecticide/miticide	Some		Modest	1.28, 1.18–1.38
Turner et al., 2010	Systematic review	4	Pesticides	Insecticides				2.14, 1.83–2.50
				Herbicides				1.73, 1.28–2.35
Van Maele-Fabry et al., 2019	Systematic review	5		Insecticides		ALL		1.28, 1.07–1.53
		3		Herbicides				1.34, 1.32–1.36
Bailey et al., 2015a	Pooled analysis	2		Insect repellent (Personal)		B-lineage ALL		1.42, 1.15–1.77
		6		Herbicide				1.34, 1.19–1.50
		3		Rodenticide				1.42, 1.17–1.73
		3		Molluscicide				1.01, 0.79–1.28
		6		Insecticide or fungicide	Some		Modest	1.26, 1.11–1.44
Bailey et al., 2015a	Pooled analysis	5	Pesticides	Pesticide used on pets	Little	B-lineage ALL	NA	1.15, 1.03–1.29
Zhou et al., 2014	Systematic review	7	Chemicals	Solvent				1.25, 1.09–1.45
		7		Petroleum	Some	ALL	NA	1.42, 1.10–1.84

* Evidence category reflects those in the same rows by exposure type; ^†^ source of prevalence is different from RR data; ^‡^ RR also includes OR.

**Table 3 cancers-14-00382-t003:** Postnatal exposure to environmental factors in relation to childhood acute lymphoblastic leukemia in their offspring, including strength of evidence, prevalence of the risk factors in Germany, and magnitude of risk (RR) with 95% confidence intervals.

Authors	Study Design	Number of Study	Exposure Group	Exposure Type/Agent	* Evidence	Leukemia/Sub Type	^†^ Prevalence	^‡^ RR 95% CI
Sun et al., 2014	Systematic review	11	Air pollution	Traffic density		Leukemia		1.03, 0.98–1.09
Filippini et al., 2019	Systematic review	16		Traffic density		Leukemia		1.09, 1.00–1.20
		9		Traffic density		ALL		1.05, 0.96–1.16
		3		Traffic density <6 years	Little	ALL	NA	1.02, 0.99–1.05
		7		Nitrogen Dioxide		Leukemia		1.04, 0.90–1.19
		4		Nitrogen Dioxide		ALL		1.02, 0.89–1.18
		2		Nitrogen Dioxide children <6 years	No	ALL	High	1.10, 0.92–1.32
Filippini et al., 2015	Systematic review	4	Chemicals	Proximity to petrol station		Leukemia		1.83, 1.42–2.36
Filippini et al., 2019	Systematic review	8		Benzene		Leukemia		1.27, 1.03–1.56
		7		Benzene		ALL		1.09, 0.88–1.36
				Benzene children < 6 years	Some	ALL	NA	1.19, 1.00–1.40
Schuz et al., 2007	Pooled analysis	4	Electromagnetic fields	ELF-MF (10:00 p.m.–6:00 a.m.) 0.1 ≤ 0.2 µT		Leukemia		1.11, 0.91–1.36
				ELF-MF (10:00 p.m.–6:00 a.m.) 0.2 ≤ 0.4 µT				1.37, 0.99–1.90
				ELF-MF (10:00 p.m.–6:00 a.m.) ≥ 0.4 µT				1.93, 1.11–3.35
				ELF-MF 24-/48-h 0.1 ≤ 0.2 µT				1.09, 0.89–1.32
				ELF-MF 24-/48-h 0.2 ≤ 0.4 µT				1.20, 0.89–1.06
				ELF-MF24-/48-h ≥ 0.4 µT				1.98, 1.18–3.35
Ahlbom et al., 2000	Pooled analysis	9		ELF-MF 0.1 ≤ 0.2 µT		Leukemia		1.08, 0.88–1.32
		9		ELF-MF 0.2 ≤ 0.4 µT				1.12, 0.84–1.51
		9		ELF-MF ≥ 0.4 µT				2.08, 1.30–3.33
		7		ELF-MF 0.1 ≤ 0.2 µT				1.07, 0.81–1.41
		7		ELF-MF 0.2 ≤ 0.3 µT				1.16, 0.69–1.93
		7		ELF-MF ≥ 0.3 µT				1.44, 0.88–2.36
Greenland et al., 2000	Pooled analysis	12		ELF-MF 0.1–0.2 µT—Wire Code Alone		Leukemia		1.02, 0.81–1.29
				ELF-MF 0.2–0.3 µT—Wire Code Alone				1.01, 0.69–1.48
				ELF-MF > 0.3 µT—Wire Code Alone				1.38, 0.89–2.13
Zhao et al., 2014	Systematic review	7		ELF-MF 0.1 ≤ 0.2 µT		ALL		1.09, 0.85–1.39
				ELF-MF 0.2 ≤ 0.4 µT		ALL		1.04, 0.73–1.48
				ELF-MF ≥ 0.4 µT		ALL		2.43, 1.30–4.55
Greenland et al., 2000	Pooled analysis	12		ELF-MF 2 µT		Leukemia		1.08, 0.86–1.35
				ELF-MF 0.2–0.3 µT				1.10, 0.76–1.60
				ELF-MF > 0.3 µT				1.52, 0.99–2.33
Amoon et al., 2021	Pooled analysis	4		ELF-MF ≥ 0.4 μT		Leukemia		1.01, 0.61–1.66
				ELF-MF 0.1 ≤ 0.2 μT				1.10, 0.80–1.53
				ELF-MF 0.2 ≤ 0.4 μT				0.75, 0.46–1.21
Seomun et al., 2021	Systematic review	27		ELF-MF 0.4µT	Some	Leukemia	Rare	1.72, 1.25–2.35
Liu et al., 2011	Systematic review	7	Lifestyle, behaviour, infection	Paternal smoking			High	1.24, 0.96–1.60
Chunxia et al., 2019	Systematic review	3		Maternal smoking			Common	0.84, 0.59–1.19
Rudant et al., 2015	Pooled analysis	11	Lifestyle, behaviour, infection	Breastfeeding		ALL		0.95, 0.89–1.02
				Breastfeeding < 6 months				1.01, 0.94–1.08
				Breastfeeding ≥ 6 months				0.86, 0.79–0.94
				Breastfeeding				0.95, 0.89–1.02
				Breastfeeding < 6 months				1.01, 0.94–1.08
				Breastfeeding ≥ 6 months				0.86, 0.79–0.94
Martin et al., 2005	Systematic review	17		Breastfeeding				0.91, 0.84–0.98
Amitay et al., 2015	Systematic review	11		Breastfeeding				0.82, 0.73–0.93
Kwan et al., 2004	Systematic review	14		Breastfeeding			High	0.76, 0.68–0.84
Urayama et al., 2010	Systematic review	9	Lifestyle, behaviour, infection	Day-care attendance any time				0.81, 0.70–0.94
		11		Day-care attendance at age ≤ 2				0.79, 0.65–0.95
Rudant et al., 2015	Pooled analysis	11		Day-care centre attendance at <1 year of age				0.77, 0.71–0.84
		11		Day-care centre attendance at <1 year of age	Some(inverse)		High	0.77, 0.71–0.84
Orsi et al., 2018	Pooled analysis	13	Lifestyle, behaviour, infection	Living on a farm	No	ALL	NA	1.09, 0.86–1.36
Orsi et al., 2018	Pooled analysis	13	Lifestyle, behaviour, infection	Contact with any pets	Some(inverse)	ALL		0.90, 0.84–0.96
Bailey et al., 2015b	Pooled analysis	4	Paint	Home paint-Any paint exposure	Some	B-lineage ALL	Rare	1.22, 1.07–1.39
Bailey et al., 2015a	Pooled analysis	5		General-Professional pest control treatments		B-lineage ALL	Modest	1.28,1.14–1.45
Van Maele-Fabry et al., 2019	Systematic review	8	Pesticides	General		ALL		1.42, 1.13–1.80
		3		General				1.24, 0.90–1.70
		3		General-Indoor exposure				1.19, 0.90–1.57
		3		General-Out door exposure				1.27, 0.93–1.72
Turner et al., 2010	Systematic review	4		General		ALL		1.40, 0.90–2.16
Turner et al., 2010	Systematic review	3		General-Indoor exposure				1.56, 1.02–2.39
Turner et al., 2010	Systematic review	4		General-Outdoor exposure				1.40, 1.05–1.87
Chen et al., 2015	Systematic review	6		General-Indoor		ALL		1.59, 1.40–1.80
Chen et al., 2015	Systematic review	6		General-Outdoor				1.15, 0.95–1.38
Chen et al., 2015	Systematic review	7		General-Indoor pesticides-professional home	Some		Modest	1.55, 1.38–1.75
Chen et al., 2015	Systematic review	7		Home pesticide				1.46, 1.29–1.65
Chen et al., 2015	Systematic review	5		Insecticides indoor				1.59, 1.39–1.81
Bailey et al., 2015a	Pooled analysis	12		Home pesticide				1.35, 1.21, 1.52
Bailey et al., 2015a	Pooled analysis	5		Household insecticide/miticide	Some	B-lineage ALL	Modest	1.23, 1.12–1.34
Vinson et al., 2011	Systematic review	20	Pesticides	Herbicides		Leukemia		1.26, 1.14–1.39
Vinson et al., 2011	Systematic review	45		Insecticides				1.17, 1.03–1.33
Bailey et al., 2015a	Pooled analysis	5		Insecticide or fungicide		B-lineage ALL		1.41, 1.26–1.59
Bailey et al., 2015a	Pooled analysis	2		Insect repellent (Personal)				1.02, 0.86–1.20
Bailey et al., 2015a	Pooled analysis	5		Herbicide				1.34, 1.21–1.48
Bailey et al., 2015a	Pooled analysis	3		Rodenticide				1.32, 1.12–1.56
Bailey et al., 2015a	Pooled analysis	3		Molluscicide		B-lineage ALL		1.06, 0.87–1.30
Chen et al., 2015	Systematic review	9		Insecticides-Outdoor				1.11, 0.60–2.05
Chen et al., 2015	Systematic review	5		Herbicides Outdoor				1.26, 1.10–1.44
Van Maele-Fabry et al., 2019	Systematic review	3		Insecticides		ALL		1.19, 0.90–1.57
Van Maele-Fabry et al., 2019	Systematic review	3		Herbicides				1.24, 0.96–1.60
Turner et al., 2010	Systematic review	3		Insecticides		ALL		1.35, 0.76–2.38
Turner et al., 2010	Systematic review	4		Herbicides	Some		Modest	0.85, 0.43–1.66
Bailey et al., 2015a	Pooled analysis	6		Pesticide used on pets		B-lineage ALL	NA	1.15, 1.03–1.29
Baker and Hoel, 2007	Systematic review	6	Radiation	Proximity to nuclear facilities Incidence All				1.25, 1.13–1.38
		6	Radiation	Proximity to nuclear facilities Incidence < 16 km				1.23, 1.07–1.40
		6	Radiation	Proximity to nuclear facilities Mortality All				1.06, 1.01–1.11
		6	Radiation	Proximity to nuclear facilities Mortality < 16 km	Some	Leukemia	Modest	1.23, 1.04–1.46
Lu et al., 2020	Systematic review	8	Radiation	Domestic radon				1.22, 1.01–1.42
		2	Radiation	Domestic radon	Conflicting	Leukemia	Moderate	0.97, 0.81–1.15
Little et al., 2018	Systematic review	7	Radiation	Low doses of ionising radiation 5–9.99 mSv		ALL		2.41, 0.64–8.65
			Radiation	Low doses of ionising radiation 10–19.99 mSv				4.45, 1.50–14.08
			Radiation	Low doses of ionising radiation 20–49.99 mSv				4.20, 1.35–13.28
			Radiation	Low doses of ionising radiation 50–100 mSv				3.97, 0.97–14.15
			Radiation	Low doses of ionising radiation RR at 100 mSv	Strong		Modest	5.66, 1.35–19.71

* Evidence category reflects those in the same rows by exposure type; ^†^ source of prevalence is different from RR data; ^‡^ RR also includes OR.

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
