# Peer review of "Environmental Risk Factors for Childhood Acute Lymphoblastic Leukemia: An Umbrella Review"

_cancers, 2022, doi:10.3390/cancers14020382_

Round 1

Reviewer 1 Report

This is a well-researched and very well-organized umbrella summary of the field of childhood leukemia (specifically the ALL subtype) and its epidemiological risk factors. The tables are meticulously and carefully constructed. The report is well written. But the rationale for this work is unclear, given the numerous and similar reviews that have already been published, as recently as 2020. The one novel contribution of the work - the description of the prevalence of these associated factors in the German population - might have been a helpful contribution, except that it was not given the same careful attention to detail as the rest of the review. The paragraph describing the results of the prevalence analysis is very brief, and lacking in detais, which is surprising giving the stated aim of the manuscript. There is no tabulation of the prevalence data, and the interpretation of each potential exposure as "rare" or not so rare is unhelpful and appears subjective.

Author Response

Response to Reviewer 1 Comments

Point 1: This is a well-researched and very well-organized umbrella summary of the field of childhood leukemia (specifically the ALL subtype) and its epidemiological risk factors. The tables are meticulously and carefully constructed. The report is well written. But the rationale for this work is unclear, given the numerous and similar reviews that have already been published, as recently as 2020.

Response 1:

The authors appreciate the time the reviewers took to review and provide useful feedback to improve the quality of the manuscript. We addressed each comment carefully and provided our responses below.

We thank the reviewer for the supportive assessment and for the constructive comments. We already noted the fact in our manuscript that there are numerous reviews and systematic reviews but no umbrella type of review yet. Hence, we stated the rationale behind our umbrella review in lines 76 -80 “As more systematic reviews and meta-analyses are published, decision-makers need to integrate the accumulating evidence for a concise evaluation to answer their questions. While systematic reviews can come to different results, umbrella reviews such as this, help to synthesize the evidence to give a consolidated overview

Based on our literature search this is the first umbrella review to synthesize environmental risk factors specific for childhood acute lymphoblastic leukemia (ALL) and furthermore structured and interpreted based on relevant exposure time windows.

Point 2: The one novel contribution of the work - the description of the prevalence of these associated factors in the German population - might have been a helpful contribution, except that it was not given the same careful attention to detail as the rest of the review. The paragraph describing the results of the prevalence analysis is very brief, and lacking in details, which is surprising giving the stated aim of the manuscript. There is no tabulation of the prevalence data, and the interpretation of each potential exposure as "rare" or not so rare is unhelpful and appears subjective.

Response 2:

We thank the reviewer for acknowledging the novelty of our umbrella review. The prevalence data used in this paper was hampered by the lack of reliability and representativeness. Also, data for some risk factors were outdated or simply not available. This was partly the reason for the brief results and discussion in the prevalence section. Categorizing the prevalence of risk factors for ALL in the German population into high (>20%), common (>10-20%), moderate (>5-10%), modest (>2-5%), and rare (<2%) was in our opinion the best approach to illustrate the available data. We have also clearly stated these short comings in the manuscript accordingly.

Reviewer 2 Report

The manuscript by Dr. Onyije and colleagues discuss the relationship between environmental factors and ALL in children. The function of the author-senior is performed by Prof. Schüz, a world-class expert in this field.

Similar narrative reviews already exist in the literature, so it is an excellent idea the authors decided to present the results in the form of an umbrella review.

I have attached notes to the manuscript below:

  1. In the introduction, the authors could mention the latest research on the relationship between childhood leukemia and iron metabolism already published in Cancers (DOI: 10.3390/cancers13123029).
  2. The aim of the work is clearly defined.
  3. I wonder if the authors could register their review in PROSPERO at this stage. However, it is not a requirement, as the study was adequately performed following PRISMA’s rules.
  4. The choice of the time (2003-2021) should be more motivated by the authors.
  5. Is the lack of restrictions as to the language of the publication, not an additional factor increasing the potential bias?
  6. Please justify why the authors searched only two bibliographic databases?
  7. I have no comments on the results - they are described in detail.
  8. Limitations should be described in more detail.
  9. Overall, exciting and very well-written paper. The manuscript is worth publishing. It has outstanding scientific and didactic values.

Author Response

Response to Reviewer 2 Comments

The manuscript by Dr. Onyije and colleagues discuss the relationship between environmental factors and ALL in children. The function of the author-senior is performed by Prof. Schüz, a world-class expert in this field.

Similar narrative reviews already exist in the literature, so it is an excellent idea the authors decided to present the results in the form of an umbrella review.

I have attached notes to the manuscript below:

Point 1: In the introduction, the authors could mention the latest research on the relationship between childhood leukemia and iron metabolism already published in Cancers (DOI: 10.3390/cancers13123029).

The authors appreciate the time the reviewers took to review and provide useful feedback to improve the quality of the manuscript. We addressed each comment carefully and provided our responses below.

Response 1:

We like to thank the reviewer for the positive assessment and the constructive comments. We have addressed the reviewer’s suggestions.

The authors have now cited the article “Unbalance in Iron Metabolism in Childhood Leukemia Converges with Treatment Intensity: Biochemical and Clinical Analysis” in line 64 as suggested.

Point 2: The aim of the work is clearly defined.

Response 2: The authors thank the reviewer for acknowledging the clarity of the aim of our umbrella review.

Point 3: I wonder if the authors could register their review in PROSPERO at this stage. However, it is not a requirement, as the study was adequately performed following PRISMA’s rules.

Response 3: The authors agree with the reviewer’s suggestion that it is not a requirement for publication. Also, PROSPERO will not accept to register a completed umbrella review, rather before the commencement of the review.

Point 4: The choice of the time (2003-2021) should be more motivated by the authors.

Response 4: This has been revised as suggested (please see lines 108-110): “The choice of this timespan was to reduce the overlap of original studies, and to better reflect current exposure circumstances in view of prevention opportunities”.

Point 5: Is the lack of restrictions as to the language of the publication, not an additional factor increasing the potential bias?

Response 5: The lack of restrictions to the language of publication rather decreased potential bias as it yielded a wider range of publications.

Point 6: Please justify why the authors searched only two bibliographic databases?

Response 6: MEDLINE (PubMed) is the most comprehensive database for health and medicine, while Web of Science (WOS) is one of the largest interdisciplinary research databases with a broad scope.

Point 7: Limitations should be described in more detail.

Response 7: Authors have made a separate paragraph starting with “weaknesses include…” and revised this section to include part of the limitation in the prevalence data, as suggested by the reviewer (Please see lines 419-420, 444-448).

Point 8: I have no comments on the results - they are described in detail. Overall, exciting and very well-written paper. The manuscript is worth publishing. It has outstanding scientific and didactic values.

Response 8

The authors deeply appreciate the reviewer for this comment.

Round 2

Reviewer 2 Report

The authors have addressed my concerns and improved the quality of the manuscript.